# The Relationship between Type 2 Diabetes, Differentiation of Self, and Emotional Distress: Jews and Arabs in Israel

**DOI:** 10.3390/nu14010039

**Published:** 2021-12-23

**Authors:** Ora Peleg

**Affiliations:** Education and School Counseling Departments, Max Stern, Yezreel Valley College, Afula 1930600, Israel; orap@yvc.ac.il

**Keywords:** T2DM, differentiation of self, ethnic differences, Jews, Arabs

## Abstract

Type 2 diabetes mellitus (T2DM) is considered a global epidemic, and is constantly on the rise. In Israel, the percentage of diabetics in the Arab population is twice that found in the Jewish population (12% and 6.2%, respectively). Findings suggest that low differentiation of self (DoS: emotional reactivity+ fusion with others, I-position, emotional cutoff) may raise vulnerability to certain physiological pathologies by increasing susceptibility to psychological distress. The major goal of this study was to test differences in DoS and emotional distress (anxiety and depressive symptoms) between diabetic and healthy participants. The second aim was to examine cultural differences within these metrics. Another purpose was to examine the relationship between DoS and emotional distress among healthy and diabetic participants. The sample included 261 participants, of whom 154 were healthy and 107 were diabetic. Diabetics reported more severe depressive symptoms, higher levels of anxiety and emotional cutoff and lower levels of I-position than healthy individuals. The groups did not differ in their levels of emotional reactivity + fusion with others. Arabs demonstrated higher levels of emotional cutoff, anxiety and depressive symptoms and lower levels of I-position than Jews. However, Arabs and Jews did not differ in their levels of emotional reactivity + fusion with others. Emotional reactivity + fusion with others contributed the most to diabetes among Arabs, while depressive symptoms contributed the most among Jews. Finally, among Jewish participants, age was positively correlated with emotional cutoff and depressive symptoms. Emotional cutoff was positively correlated with anxiety and depressive symptoms. Emotional reactivity + fusion with others was positively correlated with anxiety. Among Arab participants, age was positively correlated with emotional cutoff, anxiety and depressive symptoms. I-position was negatively correlated with all study variables. Emotional cutoff was positively correlated, anxiety and depressive symptoms. Emotional reactivity + fusion with others was positively correlated with anxiety and depressive symptoms.

## 1. Introduction

Type 2 diabetes mellitus (T2DM), a metabolic disorder characterized by high blood glucose, insulin resistance and a relative lack of insulin, is considered a global epidemic, with the number of patients estimated at about 422 million and constantly on the rise [1]. T2DM has been identified by the World Health Organization [1] as one of the four leading noncommunicable diseases contributing to premature death globally. People with diabetes are at a high risk of mortality due to physiological and psychological complications [2]. According to the International Diabetes Association, the prevalence of diabetes in Israel among individuals aged 20–79 was 9.7% [1]. In Israel, approximately 550,000 people are diabetic and another 550,000 are prediabetic [3]. T2DM has an unequal influence on ethnic/racial minorities from deprived backgrounds and occurs in complex psychosocial and cultural environments [4].

Most diabetes care and research has focused on behavioral factors such as physical activity, nutrition, smoking, and drug therapy [5] (pp. 81–89). Despite the emphasis placed on these factors, up-to-date research has revealed that certain emotional factors might also contribute to the development and severity of T2DM. Emotional factors associated with the capacity to deal with stressful situations, e.g., anxiety and depression, are of particular significance [5,6]. One possible cause of this association is the relationship between physiological stress systems and the mechanisms regulating blood glucose levels. When in a stressful state, the amygdala and hypothalamus activate hormone metabolism, affecting regulation of glucose levels including cortisol, epinephrine, noradrenaline, insulin and thyroxine [6].

The association between psychological factors (anxiety and depression) and physiological symptoms is supported by empirical evidence [7]. Studies have pointed to relations between such factors and a series of illnesses, such as cardiovascular disease [7], chronic disease and cancer [8]. Anxiety has been associated with cortisol secretion rates [9] and adrenal gland responsiveness [10]. The lower the level of anxiety, the higher the cortisol secretion rate and the better the stress regulation [9]. Research has shown that emotional factors—stress, depression, and anxiety—are increasingly associated with the rates of T2DM [9]. Moreover, adults suffering from depression have a 37–60% greater risk of developing T2DM than those who do not suffer from depression [11].

Although emotional distress is heavily affected by familial antecedents, the role of these factors in the etiology of T2DM has not been sufficiently explored [12]. One of the most interesting factors is the possible role that low differentiation of self (DoS, a family pattern) plays in T2DM, as low DoS has been significantly related to increased levels of emotional distress [6]. This family pattern describes the ability to maintain a balance between intimacy and independence in interactions with significant others, as well as between intellectual and emotional realms, when handling stressful states. Four dimensions are connected to the individual’s level of DoS: emotional reactivity, I-position, emotional cutoff and fusion with others [13]. Poorly differentiated people tend to be overwhelmed by stressful situations, have difficulty expressing their needs and desires, develop dependent relationships, and sometimes respond to this through emotional disconnectedness and detachment from others, as well as from their own feelings [13].

A growing body of empirical evidence links DoS to physical and psychological health. Studies examining the familial sources of elevated levels of anxiety, distress, and stress have found DoS to be an important pattern influencing their regulation [6,14]. DoS has been found to be positively related to psychological wellbeing [15], satisfaction and quality of life [14] and negatively related to depression [16], emotional distress [17] and anxiety [14,15]. A few studies have found adverse associations with a number of behaviors and somatic symptoms, such as fibromyalgia [18], alcoholism [19], eating disorders [20]) and T2DM [5]. These results were observed in both individualist societies and in collectivist societies, such as Arabs living in Israel as a minority [21].

### 1.1. Ethnic Differences in Behavioral, Psychological, and Familial Factors

Cultural and ethnic diversity has recently become a major focus of T2DM research [1]. The majority of people diagnosed with T2DM fall in specific minority subgroups, such as African-American, Hispanic, Asian/Pacific Islanders and American Indians [22]. In Israel, the percentage of diabetics in the Arab population is twice than in the Jewish population (12% and 6.2%, respectively). The rate in Arab men is 1.7 times higher than the rate in Jewish men, and the rate in Arab women is 2.2 times higher than the rate in Jewish women [23]. Moreover, Arab women suffer from the disease at a younger age than Jewish women [24]. Among Jews, diabetes rates are higher among men than women in all age groups, whereas for Arabs up to age 55, rates are higher for men, while for Arabs over age 55, they are higher for women.

The current study was conducted among Jewish and Arab participants living in Israel. Jewish families, which comprise the majority of residents, are usually considered Westernized and as following democratic norms and relationships, although families of Eastern origin (initially from Arab countries) are more traditional and patriarchal. The Arab population, consisting of Muslims—mainly Sunni (83.4%), including peasants, urban dwellers and Bedouin (3.5%)—as well as smaller subpopulations of Christians (8.4%) and Druze (8.2%), differ from the Jewish majority in terms of language, religion, culture, historical narrative, geographical area, education system and socioeconomic status [21,25]. Most Arab residents in Israel (21.1%) [23] live in Arab villages and towns in three main areas: The Galilee, the Triangle, and the Negev. Ten percent live in ethnically mixed cities, such as Acre, Haifa, and Jaffa. The key elements characterizing Arab culture are compliance with religious, traditional and cultural norms. Accordingly, social relationships are based on commitment and devotion to family and friends, and individuals are encouraged to uphold communal norms rather than to be independent and address their personal needs and expectations [25,26].

Although Arab society in Israel is more collectivist than the Jewish population, the former is in a period of transition between traditionalism and modernity, where an attempt has been made to mimic the norms of Western culture without really integrating them. Therefore, most Arab Muslims and Arab Christians living in Israel are considered bicultural [25,26]. The accelerated urbanization process in Arab society, and the fact that this group is a minority, is accompanied by increased levels of distress [21], as well as changes in family patterns, lifestyle and dietary habits, leading to the conversion of traditional carbohydrate-rich foods into foods rich in simple carbohydrates. These changes explain the high rates of diabetes found in the population [27]. In addition, studies comparing Jews and Arabs in Israel have indicated that the latter reported higher levels of emotional cutoff and social anxiety than the former. Gender comparisons yielded higher levels of emotional reactivity, fusion with others and anxiety among women than men [21].

### 1.2. Research Objectives and Hypotheses

This research addressed an important gap in the literature, adopting an ethnic comparative perspective to explore the involvement of DoS and emotional distress factors in T2DM. The study was conducted among two groups (diabetics and healthy participants) belonging to two Israeli cultures (Jews and Arabs) and both genders. In light of the disproportionate incidence of T2DM among ethnic minorities and specifically among Israeli Arabs, the study tested cultural and gender differences in DoS and emotional distress (anxiety and depressive symptoms) that might be associated with T2DM. This could shed light on the complexity of the antecedents that may be associated with the illness and influenced by different cultural backgrounds. Such an in-depth and comprehensive understanding may provide a profile characterizing the disease in each ethnic group.

Taken together, the study’s findings suggested that low DoS may raise vulnerability to certain physiological pathologies by increasing susceptibility to psychological distress. Thus—and given the evidence of associations between high blood glucose levels, low levels of DoS [6], and high levels of anxiety [9], and depressive symptoms [28]—it is reasonable to assume that differences will be found between diabetic individuals (diagnosed with T2DM, with fasting glucose ≥ 126 mg/dL) and healthy ones (fasting glucose levels = 70–100 mg/dL and hemoglobin A1c < 5.7%) in terms of DoS (emotional reactivity, I-position, emotional cutoff, fusion with others) and emotional distress (anxiety, depressive symptoms). We expected diabetic participants to report lower functioning levels than healthy participants in all dimensions (Hypothesis 1).

Based on documented ethnic-related variability in risk of T2DM [23], levels of emotional cutoff [21], and levels of emotional distress [6], it was deemed reasonable to assume that Arab (diabetic and healthy) participants would report lower levels of functioning in these dimensions than Jewish respondents. Therefore, we hypothesized that Arabs would report lower levels of emotional cutoff (DoS) and higher levels of anxiety and depressive symptoms (emotional distress) than Jews (Hypothesis 2). Based on documented gender-related findings on DoS and emotional distress [20], we also predicted (Hypothesis 3) that women would report higher levels of emotional reactivity and fusion with others (DoS), as well as anxiety and depressive symptoms (emotional distress), than men. Finally (Hypothesis 4), we assumed that DoS would be associated with emotional distress, high levels of emotional reactivity, emotional cutoff and fusion with others and a low level of I-position would be associated with high levels of stress and depressive symptoms.

## 2. Methods

### 2.1. Participants

The present study included 261 participants (mean age 42.9, range 26–71; 166 women and 95 men), of whom 154 (59%) were healthy and 107 (41.0%; 95% CI: 35.2–47.0) were diabetic. The healthy group was comprised of 118 (76.6%) women and 36 (23.4%) men. In terms of cultural group, 113 (73.4%) healthy participants were Jews, of whom 86 (76.1%) were women and 27 (23.9%) were men; while 41 (36.6%) were Arabs, of whom 32 (78.0%) were women and 9 (22.0%) were men. The diabetic group consisted of 59 (55.1%) men and 48 (44.9%) women. In this group, 64 (59.8%) were Arabs, comprised of 35 (54.7%) men and 29 (45.3%) women; and 43 (40.2%) were Jews, comprised of 24 (55.8%) men and 19 (44.2%) women. Index inclusion in the diabetics group on criteria were: self-reported diagnosis of T2DM; BMI 25–47 kg/m^2^ inclusive; hemoglobin A1c ≤ 11%; currently under the care of a healthcare provider. The control group was adjusted to the group of diabetics in terms of background variables but, unlike the group of diabetics examined, they were not diagnosed as diabetics according to the indices mentioned.

### 2.2. Instruments

A background questionnaire was used; it included items on diet, exercise, smoking habits, ethnicity, age, gender, religion, work status (whether employed), marital status, residence, education and onset of diabetes. Diabetic patients provided data on their blood glucose level.

Blood glycemic control levels were evaluated by a blood test taken after 12 h of fasting. A value of 70–100 mg/dL was considered normal (healthy) and 126 mg/dL and above indicated diabetes.

DoS was measured by the *Differentiation of Self Inventory-Revised* (DSI-R); [15,29], translated to Hebrew [30,31]. This is a 40-item self-report inventory, in which possible responses to each item fall along a six-point Likert scale, ranging from 1 (not at all true of me) to 6 (very true of me). The original DSI-R included four subscales assessing emotional reactivity, I-position, emotional cutoff, and fusion with others. Sample items include: “People have remarked that I am overemotional” (emotional reactivity); “No matter what happens in my life I know that I will never lose my sense of who I am” (I-position); “I tend to distance myself when people get too close to me” (emotional cutoff); “When my spouse or partner is away for too long, I feel like I am missing a part of me” (fusion with others). The current study combined the subscales of emotional reactivity and fusion with others, in light of the findings of a recent study [6] that reexamined validation of the subscale structure of the DSI-R on a sample of diabetic Jewish and Arab Israeli participants. In that study, an exploratory principal components factor analysis yielded three factors: (1) I-position (9 items), (2) emotional cutoff (12 items); and (3) items relating to both emotional reactivity and fusion with others (19 items). In the current study, internal consistency (Cronbach’s alpha) of the adapted version of the DSI-R were: 0.92 for the total score, 0.73 for I-position, 0.85 for emotional cutoff and 0.88 for emotional reactivity + fusion with others.

Levels of anxiety were assessed by the State-Trait Anxiety Inventory (STAI); [32] translated to Hebrew [33]. In this 20-item scale, respondents were asked to circle the number that best described how they usually feel. Sample item: “Some unimportant thought runs through my mind and bothers me.” Possible scores ranged from 20 to 80. In the current study, mean scores were computed, ranging from 1 to 3. Internal consistency (Cronbach’s alpha) in the present study was 0.92.

Depressive symptoms were measured by the Beck Depression Inventory (BDI) [34] translated to Hebrew [35]. This 21-item inventory assessed the level of depression among adolescents and adults (ages 13–80) over the past week. Sample item: “1. I’m not sad, 2. I’m sad, 3. I’m sad all the time and can’t get rid of it, 4. I’m so sad or unhappy that I can’t stand it.” Respondents were asked to mark each statement group with the best description of their feelings during the past week. Scores were categorized as follows: 0–9 = normal and without symptoms, 10–18 = mild depression, 19–29 = moderate depression, 30–63 = severe depression. Internal consistency (Cronbach’s alpha) in the current study was 0.93.

Translation procedure: all the above instruments were translated to Arabic for the purpose of this research. The translation process followed guidelines for cultural adaptation in order to assure content validity. Two independent translators, fluent in both English and Arabic and knowledgeable in psychology and education, translated the inventories. Backwards translation (from Arabic to English) was carried out by two native Arabic speakers, fluent in English, who had no prior knowledge of the instrument. The translations were compared and differences were discussed within the translation team to reach consensus. During the translation process, the overall aim was to ensure comprehensibility and capture the original idea of each item.

### 2.3. Procedure

The complete study protocol was approved by the Max Stern Yezreel Valley College Institutional Review Board. Some participants were reached in medical centers and health clinics. Others were recruited by the snowball sampling method; that is, a small number of individuals were recruited for the first circle of participants, who then referred others, and so on. All participants signed an informed consent form. Blood glucose data was based on the last blood test taken over the past two months, the results of which participants were asked to bring to the intake meeting. Completion of the questionnaires was voluntary, and respondents were told that they could stop participation at any point. All were assured of anonymity and discretion.

### 2.4. Data Analyses

Prevalence of diabetes was computed for each ethnic group and each gender. Means, standard deviations, and the distribution of the study variables were calculated. MANCOVA of the DoS dimensions (I-position, emotional cutoff, emotional reactivity + fusion) was run, with age as covariate, to assess ethnic group differences between these variables. ANCOVA was performed to assess ethnic group differences in anxiety and depressive symptoms. Pearson correlations were calculated by ethnic group, and by ethnic group and diabetes status, and differences in correlation coefficients were tested via Fisher Z-transformations. A logistic regression was then performed for each ethnic group using the DoS dimensions, anxiety, and depressive symptoms as predictors of T2DM in both cultural groups, with age and gender as covariates. Finally, a stepwise logistic regression was run to ascertain significant predictors.

## 3. Results

To calculate sample size, G-Power was used: For an effect size of 0.025, alpha = 0.05 and power of 80% a sample size of 258 was needed to run the MANCOVA. For a medium effect size (f = 0.25) a sample size of 237 at 80% power was needed to carry out ANCOVA with one covariate, 2 groups and the 3 main effects plus all 2-way interactions (for the anxiety and depression variables).

Overall, 41.0% of participants (95% CI: 35.2–47.0) were diabetic. The prevalence of T2DM was significantly different between the two ethnic groups, with 27.6% (43 of the Jewish participants diabetic compared to 60.9 (64) of the Arab participants (χ^2^ = 26.55, *p* < 0.001). In addition, the prevalence of diabetes was significantly higher among male participants than females (55.1% vs. 44.9%, χ^2^ = 28.95, *p* < 0.001); this was true within each ethnic group.

Table 1 presents means, standard deviations, and the distribution of the study variables. Data (skewness and kurtosis) indicated that the variables were approximately bell-shaped, allowing for parametric analysis. Table 2 presents the study variables by ethnic group, study group and gender.

MANCOVA of the DoS dimensions (I-position, emotional cutoff, emotional reactivity + fusion), with age as covariate, revealed statistically significant differences between diabetic and healthy individuals [F(3,255) = 6.60, *p* < 0.001, partial eta squared = 0.072]. Specifically, individuals with T2DM reported lower levels of I-position [F(1,257) = 7.15, *p* < 0.001, partial eta squared = 0.027] and higher levels of emotional cutoff [F(1,257) = 15.53, *p* < 0.001, partial eta squared = 0.057] than their healthy counterparts. There were no differences in reported emotional reactivity + fusion [F(1,257) = 1.12, *p* > 0.29, partial eta squared = 0.004]. An ANCOVA of anxiety, with age as covariate, revealed that diabetic individuals reported higher levels of anxiety [F(1,257) = 6.71, *p* < 0.01, partial eta squared = 0.025] and depressive symptoms [F(1,257) = 72.00, *p* < 0.001, partial eta squared = 0.219] than their healthy counterparts.

Table 3 shows the results of the MANCOVA for the DoS dimensions, with main effects of study group, ethnic group and gender, and all 2-way interactions between them, using age as covariate. Results revealed the main effects of ethnic group for the total score of DoS [F(3, 251) = 7.08, *p* < 0.001, partial eta squared = 0.078] and gender [F(3, 251) = 10.52, *p* < 0.001, partial eta squared = 0.112], as well as interaction effects of study group by ethnic group [F(3, 251) = 3.17, *p* < 0.025, partial eta squared = 0.036] and of ethnic group by gender [F(3, 251) = 4.05, *p* < 0.008, partial eta squared = 0.046].

Specifically, there were significant study group main effects for emotional cutoff [F(1, 253) = 3.89, *p* < 0.05; partial eta squared = 0.015], with diabetic individuals reporting higher levels of this dimension than their healthy counterparts. There were also significant ethnic group main effects for I-position [F(1, 253) = 7.41, *p* < 0.007; partial eta squared = 0.028] and emotional cutoff [F(1, 253) = 12.88, *p* < 0.001; partial eta squared = 0.048], with Arab participants reporting lower levels of I-position and higher levels of emotional cutoff than their Jewish counterparts.

With respect to gender, female participants reported higher levels of emotional reactivity + fusion with others than their male counterparts [F(1, 253) = 20.02, *p* < 0.001; partial eta squared = 0.073]. Moreover, a significant ethnic group and gender interaction was found for the emotional cutoff dimension [F(1, 253) = 8.61, *p* < 0.004; partial eta squared = 0.033]. Post hoc analysis of the interaction revealed that, among Jewish participants, males reported significantly higher emotional cutoff [F(1, 152) = 8.65, *p* < 0.004, partial eta squared = 0.054] than females, while there were no gender differences among Arab participants [F(1, 103) = 1.06, *p* > 0.35, partial eta squared = 0.01].

Significant study groups by ethnic group interactions were found for I-position [F(1, 253) = 6.85, *p* < 0.009, partial eta squared = 0.026] as well as emotional reactivity + fusion with others [F(1, 253) = 6.95, *p* < 0.009, partial eta squared = 0.027]. Post hoc analysis of the interaction revealed a statistically significant difference in I-position [F(1, 103) = 11.76, *p* < 0.001, partial eta squared = 0.095] and in emotional reactivity + fusion with others [F(1, 103) = 4.97, *p* < 0.03, partial eta squared = 0.043] between diabetic Arabs and diabetic Jews, but no difference in these variables between healthy Arabs and Jews [F(1, 149) = 0.03, *p* > 0.87, partial eta squared = 0.000 for I-position; F(1, 149) = 0.01, *p* > 0.92, partial eta squared = 0.015 for emotional reactivity + fusion with others].

Table 3 also presents the results of ANCOVA for anxiety and depressive symptoms (i.e., emotional distress), with study group, ethnic group and gender as the main effects, and all 2-way interactions between variables, using age as covariate. Results for anxiety revealed significant ethnic group and gender main effects [F(1, 253) = 13.91, *p* < 0.001, partial eta squared = 0.052, F(1, 253) = 5.93, *p* < 0.02, partial eta = 0.023, respectively]. Healthy participants had significantly lower anxiety than diabetics, and women had significantly higher anxiety than men. The ANCOVA of depressive symptoms revealed a significant main effect of study group [F(1, 253) = 23.08, *p* < 0.001, partial eta squared = 0.084] and of ethnic group [F(1, 253) = 25.32, *p* < 0.001, partial eta squared = 0.091]; the diabetic group showed statistically significantly greater depressive symptoms as compared to the healthy group and the Arab group having statistically significantly greater depressive symptoms as compared to the Jewish group. There were no statistically significant interactions for either emotional distress variable.

Table 4 presents Pearson correlations between the study variables by ethnic group and study group. Among Jewish participants, age was positively correlated with emotional cutoff and depressive symptoms. Emotional cutoff was positively correlated with anxiety and depressive symptoms. Emotional reactivity + fusion with others was positively correlated with anxiety. For healthy Jewish participants, I-position was negatively correlated with all study variables. emotional reactivity + fusion was positively correlated with anxiety. For diabetic Jewish participants, emotional cutoff values were positively correlated with anxiety. There was a statistically significant difference between healthy and diabetic Jewish participants in the correlation between age and emotional cutoff (Z = −3.25, *p* < 0.001) and age and anxiety (Z = −1.90, *p* < 0.03), The results showed that these correlations were higher among diabetics; specifically, as they get older, they have higher levels of emotional cutoff and anxiety.

Among Arab participants, age was positively correlated with emotional cutoff, anxiety and depressive symptoms. I-position was negatively correlated with all study variables. Emotional cutoff was positively correlated with anxiety and depressive symptoms. Emotional reactivity + fusion with others was positively correlated with anxiety and depressive symptoms. Among healthy Arab participants, emotional reactivity + fusion with others was positively correlated with anxiety. Among diabetic Arab participants, I-position was negatively correlated with all study variables, emotional cutoff was positively correlated with anxiety and depressive symptoms, emotional reactivity + fusion was positively correlated with anxiety and depressive symptoms, and anxiety was positively correlated with depressive symptoms. There was a statistically significant difference between healthy and diabetic Arab participants in the correlations between emotional cutoff and emotional reactivity + fusion with others (Z = −2.196, *p* < 0.01), emotional cutoff and depressive symptoms (Z = −2.38, *p* < 0.009), emotional reactivity + fusion and depressive symptoms (Z = −4.39, *p* < 0.001), and anxiety and depressive symptoms (Z = −3.19, *p* < 0.001), with correlations higher among diabetic Arabs than healthy ones.

In light of the above results, two logistic regressions were performed, including all DoS dimensions as well as anxiety and depressive symptoms, to predict diabetes—one for each ethnic group, correcting for age and gender (Table 5). Among Jewish participants, only depressive symptoms were a statistically significant predictor of diabetes (χ^2^ = 12.22, *p* < 0.001; 0.5 point increase OR: 4.26, 95% CI: 1.97–9.17). No statistically significant predictor of diabetes was found for Arabs, although stepwise regression revealed emotional reactivity + fusion with others to be a significant predictor (χ^2^ = 4.77, *p* < 0.03; 1-point increase OR: 2.39; 95% CI: 1.09–5.21).

## 4. Discussion

This study presented a preliminary analysis of possible cultural differences in T2DM, DoS (I-position, emotional cutoff, and emotional reactivity + fusion with others), and emotional distress (anxiety and depressive symptoms). The major goal of the study was to examine differences in Dos and emotional distress between diabetic and healthy participants. The second aim was to explore cultural and gender differences in these indices. Another purpose was to test the association between DoS and emotional distress among healthy and diabetic participants.

The first hypothesis, which assumed that diabetic participants would report lower levels of functioning in all dimensions than healthy participants, was partially corroborated. Diabetics reported more severe depressive symptoms and higher levels of anxiety than healthy participants. This was in keeping with findings that suggested that the link between emotional distress and diabetes could be explained by the dysfunction of the hypothalamic-pituitary-adrenal (HPA) axis, which is triggered in response to stress [36]. Specifically, depressive symptoms were found to be related to increased secretion of cortisol (a glucocorticoid), the end product of HPA axis stimulation [37,38,39]. In order to increase energy availability over a short while, cortisol impedes insulin secretion and increases hepatic glucose output. An environment of persistent glucocorticoid exposure, i.e., chronic stress, exerts diabetogenic results by interfering with insulin action [40,41]. Thus, emotional distress may increase the risk of T2DM through the hyper-stimulation of cortisol and additional components of the physical tension response.

Regarding DoS, diabetics reported higher levels of emotional cutoff and lower levels of I-position than healthy individuals, but the groups did not differ in their levels of emotional reactivity + fusion with others. These results partially supported findings which showed that individual differences in susceptibility to depressive symptoms, and thus (apparently) to T2DM, probably depend on psychological characteristics, such as DoS [6]. DoS taps one’s ability to successfully balance between emotional and intellectual functioning and between intimacy and autonomy. Given that poorly differentiated people find it difficult to maintain a solid sense of self and to use rational thinking when they deal with stressful situations, it has been suggested that individuals with T2DM are likely to have greater difficulties preserving a clearly defined sense of self and sticking to their personal beliefs rather than following others’ expectations. Furthermore, they tend to isolate themselves from others and cut themselves off from their emotions when they experience stressful interpersonal situations.

Regarding hypothesis 1, Arabs demonstrated higher levels of emotional cutoff and lower levels of I-position than Jews. Here, too, the groups did not differ in their levels of emotional reactivity + fusion with others, partially confirming Hypothesis 2. Moreover, diabetic Arabs reported higher levels of emotional cutoff and emotional reactivity + fusion with others than Jews. This finding reinforced previous studies reporting higher levels of emotional cutoff among individuals from collectivist societies (Chung & Gale, 2006), especially Arabs in Israel [6,21]. A possible explanation for this is that Arabs find it difficult to reveal their emotions in public or in front of significant others due to family and cultural norms. In Arab society in general and among Muslims in particular, the expression of emotions, especially by men, is still considered a weakness, and thus Arabs (particularly men) are encouraged to suppress their feelings. The pattern of repressing emotions may impair expression of one’s true self and increase anxiety and depressive symptoms, which in turn may increase the risk of T2DM.

The examination of emotional distress indicated that Arabs demonstrated higher levels of anxiety and depressive symptoms than Jews. Moreover, diabetics in both ethnic groups reported higher levels of these two dimensions than healthy participants. These findings supported studies indicating that Arabs reported higher levels of anxiety [6] and depression [42] than Jews, and that people living in non-Western societies reported higher levels of anxiety and depression than individuals living in a Western society [43].

Arab society in Israel is a collectivist minority that encourages social solidarity and preserves relations at the expense of the individual. People in this society tend to raise physical complaints (e.g., blood glucose levels, sleep disorders and eating disorders) rather than emotional ones, such as anxiety, depression, and mental stress [44]. Furthermore, by virtue of belonging to a minority group, they are likely to experience more difficulties than the majority group in multiple domains, i.e., economic, professional, social, and emotional. Such concerns may enhance emotional distress, which in turn may increase the risk of T2DM.

These results deepened knowledge of factors that may increase the risk of T2DM and about the higher prevalence of diabetes in Arab society in Israel. Previous studies have reported that the causes include obesity, family history, inactivity, education and income. Studies have pointed out that T2DM is a result of a combination of sociodemographic factors and health behaviors [45]. Additionally, it was reported that greater risk of diabetes among Arabs were due to lifestyle factors, family history of diabetes and, among women, history of gestational diabetes [46]. The present study’s results suggest that specific personality factors may increase the risk of diabetes in the general population, and specifically in Arab society in Israel.

The present study also demonstrated gender differences, partially supporting the third hypothesis. Women reported higher levels of emotional reactivity + fusion with others than men, as was found in previous studies [47]. This may be due to overly high expectations that they function as spouses, mothers and career wives [20]. Furthermore, due to higher levels of oxytocin, estrogen and progesterone, women tend to be more sensitive and emotional than men [48]. Additionally, unlike men, women can usually express and reveal emotions without being seen as weak [49]. Interestingly, a significant gender difference was revealed among Jews in terms of levels of emotional cutoff, suggesting that Jewish men tend to be more detached than Jewish women, whereas no such difference was found among Arabs. This finding raises the need to further examine personal patterns and gender differences in non-Western minority societies.

The fourth hypothesis assumed associations between DoS and emotional distress (anxiety and depression). These relationships were examined separately for Jews and Arabs. Results revealed that emotional reactivity + fusion with others contributed the most to diabetes among Arabs, while depressive symptoms contributed the most among Jews. For Arabs, this may be connected to symbiotic relationships with families of origin and nuclear families (fusion with others) as well as a tendency to be emotionally overloaded (emotional reactivity). The relatively low DoS of Arabs in Israel compared to their Jewish counterparts may enhance emotional distress and consequently increase the risk of T2DM. For Jews, in contrast, it appears to be depression that is associated with the disease, suggesting that it would be worthwhile to continue researching the possible risk factors for depression among Jews in order to lower the risk of diabetes.

### 4.1. Limitations

The present study had a few limitations. First, given the correlational nature of the study, interpretation of causality should be made with caution, as the mere diagnosis of T2DM could cause increased arousal of anxiety and depression. Longitudinal studies are needed to provide more extensive data on the suggested causal relationships between T2DM and cultural, familial, and psychological dimensions. Second, the current study did not examine the possibility that Arabs and Jews in Israel have a different lifestyle that might be related to high blood glucose levels via an increased tendency for unhealthy behaviors, such as smoking, improper diet, and avoiding exercise. These factors will be the subject of a follow-up study. Finally, due to the difficulty in obtaining the cooperation of people with T2DM, a convenience sample was used, which may have impaired efforts to generalize results to the general population of diabetics.

### 4.2. Conclusions and Contributions

Notwithstanding these limitations, the research findings make a number of theoretical and applied contributions. Theoretically, the study presents a new perspective on risk factors for T2DM and the likelihood of cultural, gender and psychological dimensions being involved in the etiology of the disease. The outcomes significantly deepened our understanding of the precise cultural and psychological processes that increase risk of T2DM. Specifically, while the relationship between depression and the illness has been well documented, the current study was the first to point to poor DoS as another psychological construct that may increase vulnerability to the disease among Jews and Arabs alike. The results suggest a model whereby diabetics experience higher levels of emotional distress (anxiety and depression) and lower levels of DoS than healthy people. Finally, the results also point to cultural differences in vulnerability to the disease.

### 4.3. Practical Implications

The study results also have practical implications. They suggest that psychological interventions for the treatment and prevention of T2DM should be specially tailored to address the specific difficulties in each culture. Diabetic Arabs of both genders had lower levels of DoS, suggesting that sharing feeling and thoughts may lower their risk of developing T2DM, while among Jews, treatment of depression may lower the risk. These findings could also raise awareness among medical staff and psychologists about familial and psychological factors that increase the risk of diabetes. For instance, physicians may recommend that individuals or families engage in emotional therapy aimed at changing personal patterns in order to improve management of T2DM and prevent deterioration of the disease.

## Figures and Tables

**Table 1 nutrients-14-00039-t001:** Means, SD, and ranges of differentiation of self subscales, anxiety and depressive symptoms by study and ethnic groups.

		Total (*n* = 261)					Healthy (*n* = 154)			Diabetic (*n* = 107)		
Variable		Mean	SD	Range	SK	K	Mean	SD	Range	Mean	SD	Range
Differentiation of self												
IP	All	4.18	0.84	1.00–6.00	−0.52	0.39	4.29	0.70	2.22–5.89	4.01	0.99	1.00–6.00
	Jewish (*n* = 156)	4.32	0.73	2.22–6.00	−0.31	−0.18	4.30 (*n* = 113)	0.70	2.22–5.67	4.40 (*n* = 43)	0.80	2.89–6.00
	Arab (*n* = 105)	3.95	0.94	1.00–5.89	−0.42	0.17	4.27 (*n* = 41)	0.72	2.33–5.89	3.75 (*n* = 64)	1.02	1.00–5.56
EC	All	2.76	1.00	1.00–5.58	0.53	−0.14	2.47	0.80	1.00–4.25	3.19	1.10	1.25–5.58
	Jewish (*n* = 156)	2.50	0.84	1.00–5.42	0.39	−0.23	2.38 (*n* = 113)	0.80	1.00–4.25	2.79 (*n* = 43)	0.89	1.42–5.42
	Arab (*n* = 105)	3.16	1.08	1.25–5.58	0.32	−0.65	2.70 (*n* = 41)	0.76	1.33–4.17	3.45 (*n* = 64)	1.15	1.25–5.58
ER + FO	All	3.50	0.91	1.37–5.67	−0.11	−0.52	3.45	0.89	1.37–5.67	3.58	0.93	1.42–5.58
	Jewish (*n* = 156)	3.46	0.88	1.42–5.67	0.09	−0.32	3.52 (*n* = 113)	0.90	1.42–5.67	3.31 (*n* = 43)	0.82	1.42–5.26
	Arab (*n* = 105)	3.57	0.95	1.37–5.58	−0.16	−0.69	3.26 (*n* = 41)	0.84	1.37–4.84	3.76 (*n* = 64)	0.97	1.58–5.58
Anxiety	All	1.93	0.58	1.00–3.85	0.90	0.50	1.80	0.44	1.00–3.85	2.11	0.71	1.00–3.85
	Jewish (*n* = 156)	1.76	0.45	1.00–3.75	1.08	2.13	1.77	0.41	1.05–3.05	1.76	0.54	1.00–3.75
	Arab (*n* = 105)	2.18	0.67	1.00–3.85	0.38	−0.63	1.91	0.51	1.00–3.85	2.35	0.71	1.05–3.85
Depressive symptoms	All	1.45	0.52	0.95–3.32	1.42	1.44	1.20	0.22	0.95–2.00	1.82	0.59	0.95–3.32
	Jewish (*n* = 156)	1.28	0.35	0.95–2.68	1.67	2.55	1.16	0.19	0.95–1.82	1.60	0.46	0.95–2.68
	Arab (*n* = 105)	1.70	0.62	0.95–3.32	0.77	−0.36	1.29	0.28	0.95–2.00	1.96	0.64	0.95–3.32

Note: SK = Skewness. K = Kurtosis. IP = I-position. EC = Emotional cutoff. ER + FO = Emotional reactivity and fusion with others.

**Table 2 nutrients-14-00039-t002:** Means (SD) of differentiation of self subscales, anxiety and depressive symptoms by ethnic group, study group and gender.

	Jewish				Arab			
	Healthy		Diabetic		Healthy		Diabetic	
	Male (*n* = 27)	Female (*n* = 86)	Male (*n* = 24)	Female (*n* = 19)	Male (*n* = 9)	Female (*n* = 32)	Male (*n* = 35)	Female (*n* = 29)
IP	4.58 (0.55)	4.21 (0.72)	4.44 (0.84)	4.36 (0.78)	4.48 (0.78)	4.21 (0.70)	3.62 (1.08)	3.92 (0.93)
EC	2.58 (0.83)	2.32 (0.79)	3.12 (0.85)	2.37 (0.78)	2.40 (0.57)	2.79 (0.79)	3.33 (1.21)	3.61 (1.07)
ER + FO	3.12 (0.82)	3.64 (0.89)	3.20 (0.82)	3.45 (0.81)	2.84 (0.82)	3.38 (0.82)	3.62 (0.98)	3.93 (0.95)
Anxiety	1.60 (0.30)	1.82 (0.43)	1.81 (0.61)	1.70 (0.46)	1.79 (0.35)	1.94 (0.55)	2.30 (0.75)	2.41 (0.65)
Depressive symptoms	1.10 (0.12)	1.18 (0.20)	1.57 (0.44)	1.64 (0.49)	1.13 (0.13)	1.34 (0.29)	1.94 (0.65)	1.98 (0.63)

Note: IP = I-position. EC = Emotional cutoff. ER + FO = Emotional reactivity and fusion with others.

**Table 3 nutrients-14-00039-t003:** F tests from MANCOVA of the differentiation of self variables and ANCOVA of anxiety and depressive symptoms, with age as covariate.

	Study Group			Ethnic Group			Gender			Study Group by Ethnic Group			Study Group by Gender			Ethnic Group by Gender		
	F(1,253)	*p*	*ἠ* ^2^	F(1,253)	*p*	*ἠ* ^2^	F(1,253)	*p*	*ἠ* ^2^	F(1,253)	*p*	*ἠ* ^2^	F(1,253)	*p*	*ἠ* ^2^	F(1,253)	*p*	*ἠ* ^2^
IP	3.72	0.06	0.014	7.41	0.007	0.028	3.08	0.08	0.012	6.85	0.009	0.026	1.25	0.26	0.005	0.15	0.70	0.001
EC	3.89	0.05	0.015	12.88	0.001	0.048	0.15	0.70	0.001	1.02	0.322	0.004	0.87	0.35	0.003	8.61	0.004	0.033
ER + FO	1.95	0.16	0.008	0.52	0.47	0.002	20.02	0.001	0.073	6.95	0.009	0.027	0.66	0.42	0.003	0.02	0.88	0.000
Anxiety	2.02	0.16	0.008	13.91	0.001	0.052	5.93	0.02	0.023	0.18	0.67	0.001	0.69	0.41	0.003	0.71	0.40	0.003
Depressive symptoms	23.08	0.001	0.084	25.32	0.001	0.091	1.24	0.27	0.005	1.43	0.23	0.006	1.94	0.16	0.008	0.11	0.92	0.000

Note: *ἠ*^2^ = partial eta squared. IP = I-position. EC = Emotional cutoff. ER + FO = Emotional reactivity and fusion with others.

**Table 4 nutrients-14-00039-t004:** Pearson correlations between the study variables by ethnic and study groups.

	Jewish					Arab				
	IP	EC	ER + FO	Anxiety	Depressive Symptoms	IP	EC	ER + FO	Anxiety	Depressive Symptoms
Age Healthy Diabetic *p*	0.109 0.053 0.117 0.36	0.269 ** −0.142 0.411 ** 0.001	−0.020 0.000 0.149 0.20	0.157 −0.085 0.248 0.03	0.423 ** 0.036 0.103 0.35	−0.242 −0.099 −0.066 0.44	0.269 * 0.209 0.105 0.30	0.204 −0.042 0.081 0.28	0.252 * 0.044 0.194 0.23	0.376 ** 0.266 0.052 0.14
IP Healthy Diabetic *p*	-	−0.180 −0.249 * −0.153 0.29	−0.470 ** −0.555 ** −0.304 0.04	−0.188 −0.349 ** 0.003 0.02	−0.116 −0.293 * −0.090 0.12	-	−0.436 ** −0.156 −0.463 ** 0.06	−0.598 ** −0.442 * −0.619 ** 0.12	−0.332 ** −0.256 −0.308 * 0.39	−0.524 ** −0.281 −0.532 ** 0.07
EC Healthy Diabetic *p*	-	-	0.361 ** 0.385 ** 0.439 * 0.36	0.325 ** 0.242 0.370 * 0.21	0.343 ** 0.232 0.254 0.45	-	-	0.668 ** 0.422 * 0.720 ** 0.01	0.528 ** 0.305 0.551 ** 0.07	0.528 ** 0.151 0.571 ** 0.009
ER + FO Healthy Diabetic *p*	-	-	-	0.477 ** 0.515 ** 0.482 ** 0.40	0.174 0.360 ** 0.175 0.13	-	-	-	0.502 ** 0.536 ** 0.464 ** 0.32	0.504 ** −0.114 0.664 ** 0.001
Anxiety Healthy Diabetic *p*	-	-	-	-	0.257 ** 0.370 ** 0.081 0.04	-	-	-	-	0.501 ** 0.023 0.597 ** 0.001

* *p* < 0.01 ** *p* < 0.001. Note: IP = I-position. EC = Emotional cutoff. ER + FO = Emotional reactivity and fusion with others.

**Table 5 nutrients-14-00039-t005:** Logistic regression for the prediction of diabetes by ethnic group.

	Jewish						Arab					
		All Variables			Stepwise		All Variables			Stepwise		
	OR	95% CI	*p*	OR	95% CI	*p*	OR	95% CI	*p*	OR	95% CI	*p*
Age	1.20	1.12–1.28	0.001	1.19	1.12–1.27	<0.001	1.30	1.16–1.45	0.001	1.31	1.18–1.44	<0.001
Male	1.92	0.51–7.17	0.33	2.62	0.82–8.40	0.11	8.23	1.64–41.13	0.01	7.66	1.68–34.98	0.009
IP	1.02	0.39–2.62	0.98	-	-	-	1.16	0.32–4.13	0.82	-	-	-
EC	1.16	0.48–2.81	0.74	-	-	-	1.02	0.37–2.83	0.96	-	-	-
ER + FO	0.54	0.24–1.22	0.14	-	-	-	2.59	0.73–9.09	0.14	2.39	1.09–5.21	0.03
Anxiety	1.58	0.09–27.78	0.75	-	-	-	0.26	0.01–9.71	0.47	-	-	-
Depressive symptoms	20.40	3.74–111.11	<0.001	18.18	3.89–83.33	<0.001	3.05	0.65–14.28	0.16	-	-	-

Note: IP = I-position. EC = Emotional cutoff. ER + FO = Emotional reactivity and fusion with others.

## Data Availability

All data is stored digitally by the researcher.

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
