# Peer review of "The Relationship between Type 2 Diabetes, Differentiation of Self, and Emotional Distress: Jews and Arabs in Israel"

_nutrients, 2021, doi:10.3390/nu14010039_

Round 1

Reviewer 1 Report

The manuscript submitted to Nutrients by Peleg Ora titled: "The relationship between type 2 diabetes, differentiation of self, and emotional distress: Jews and Arabs in Israel", is a an interesting study aiming at investigating the different approaches and takes of Jews and Arabs in Israel towards T2DM. The study is interesting with good organization and structure and provides a good amount of information.  

The reviewer would like to offer a few points for the improvement of the manuscript. 

  1. In terms of population size was there a power calculation to determine optimal size?
  2. Please consider including a section at the end of the discussion section stating/discussing strengths and limitations of the study?
  3. What were the confounding factors considered and how were they normalized/accounted for?
  4. BMI does not have units as it is an index. Please remove units.
  5. For the discussion section: Is there literature pertinent to illustrating differences that could explain/influence T2DM outcomes between Jewish and Arab populations? 
  6. Please include affiliation(s) and information of the author.

Nice work overall.

Author Response

Dear Reviewer,

Enclosed please find my revised paper, “The relationship between Type 2 Diabetes, differentiation of self, and emotional distress: Jews and Arabs in Israel"" which I am resubmitting for review. I would like to take this opportunity to thank you and the reviewers for their helpful comments. Below is a brief description of the main revisions to the paper.

Comment: In terms of population size was there a power calculation to determine optimal size?

Answer:  Thanks to the reviewer's comment the following paragraph is now written in the Results chapter:

"For an effect size of  0.025, alpha=.05 and power of 80% a sample size of 258 is needed to run the MANCOVA . For a medium effect size (f=0.25) a sample size of 237 at 80% power is neeed to carry out ANCOVA with 1 covariate, 2 groups and the 3 main effects plus all 2- way interactions (for the anxiety and depression variables)".

Comment: Please consider including a section at the end of the discussion section stating/discussing strengths and limitations of the study?

Answer: These paragraphs are written at the end of the discussion. Headlines have been added to highlight them.

Comment: What were the confounding factors considered and how were they normalized/accounted for?

Answer: I controlled for age in all the analyses. BMI was considered as a confounder.

Comment: BMI does not have units as it is an index. Please remove units.

Answer: BMI is not a unitless measure.  BMI is the patients' weight in kg divided by his/her height squared and thus does have units. 

Comment: For the discussion section: Is there literature pertinent to illustrating differences that could explain/influence T2DM outcomes between Jewish and Arab populations? 

Answer: Explanations based on studies, examining factors that increase the risk of T2DM in the Arab population, have been added to the discussion.

Comment: Please include affiliation(s) and information of the author.

Answer: All the author's details are listed on the title page

Comment: Nice work overall.

Answer: Thanks.

Sincerely yours

Reviewer 2 Report

Dear Authors:

Regarding the manuscript with title “The relationship between type 2 diabetes, differentiation of self, and emotional distress: Jews and Arabs in Israel”, I have some major and minor comments to address.

Comment 1:

On Abstract only the two first hypotheses described by authors are answered on Results (“Diabetics reported more severe depressive symptoms, higher levels of anxiety and emotional cutoff and lower levels of I-position than healthy individuals. The groups did not differ in their levels of emotional reactivity+fusion with others. Arabs demonstrated higher levels of emotional cutoff, anxiety and depressive symptoms and lower levels of I-position than Jews. However, Arabs and Jews did not differ in their levels of emotional reactivity+fusion with others were related to the hypotheses assumed by authors”). No results related to hypotheses number 3 and 4 are presented on Abstract

Comment 2:

Lines 367-370: The sentence “The present study presents a preliminary analysis of possible cultural differences in T2DM, DoS (I-position, emotional cutoff, and emotional reactivity+fusion with others), and emotional distress (anxiety and depressive symptoms). The aim was to investigate whether cultural variation in DoS and emotional distress indices may increase the risk of developing diabetes” was not related to the 4 hypotheses authors stated on Abstract.

Comment 3:

Lines 134-137: “In light of the disproportionate incidence of T2DM among ethnic minorities and specifically among Israeli Arabs, the study tests cultural and gender differences in DoS and emotional distress (anxiety and depressive symptoms) that might be associated with T2DM.”

Taken into account the previous sentence, why hypotheses 2 and 3 focused on all participants instead of diabetes patients only?

Comment 4:

On lines 444-446: “The fourth htpothesis assumed associations between DoS and emotional distress (anxiety and depression). We looked at this issue comparing between ethnic groups in the indices that predict T2DM”. I can not understand the relation between the two previous sentences. Associations between DoS and emotional distress is different from the comparison between ethnic groups of the índices of Dos and emotional distress that can predict T2DM!!

Minor Comments:

Comment 1:

On Abstract, authors must add the purpose of the study before presenting the hypotheses. If the word limit does not allow i tis preferable to add the purpose of the study instead of the hypotheses.

Comment 2:

Differentiation of self is not the same concept as emotional cutoff and on Abstract both concepts are described as DoS.

Comment 3:

References are not according to authors’guidelines. Authors must check it. In the text, reference numbers should be placed in square brackets [ ]; References must be numbered in order of appearance in the text (including table captions and figure legends) and listed individually at the end of the manuscript.

Comment 4:

Regarding authors’guidelines, authors must also chack the type and size of letter. For example, the letter on Bibliography is of a different size of the rest of the paper

Commet 5:

On line 40, authors must change “sports” by “physical activity”

Comment 6:

On line 48, authors must change “bodily glucose levels” by “glucose levels”

Comment 7:

Lines 185-186: Authors must change “glycemic control levels” by “blood glucose levels”.

Another question I have: individuals with values between 100 mg/dl and 126 mg/dl are considered healthy or with diabetes?

Comment 8:

On line 245, authors stated that “ANCOVA was performed to assess ethnic group differences in anxiety and depressive symptoms.” As ANCOVA evaluates whether the means of a dependent variable are equal across levels of a categorical independent variable, I question authors if they use ANCOVA or MANCOVA to assess ethnic group differences in anxiety and depressive symptoms?

Comment 9:

On line 402: Authors must change Looking at the two ethnic groups, the first hypothesis was corroborated. Arabs by “Regarding Hypothesis 2, Arabs”

Author Response

December 19, 2021

Dear Reviewer,

Enclosed please find my revised paper, “The relationship between Type 2 Diabetes, differentiation of self, and emotional distress: Jews and Arabs in Israel"" which I am resubmitting for review. I would like to take this opportunity to thank you and the reviewers for their helpful comments. Below is a brief description of the main revisions to the paper.

Comment: On Abstract only the two first hypotheses described by authors are answered on Results (“Diabetics reported more severe depressive symptoms, higher levels of anxiety and emotional cutoff and lower levels of I-position than healthy individuals. The groups did not differ in their levels of emotional reactivity+fusion with others. Arabs demonstrated higher levels of emotional cutoff, anxiety and depressive symptoms and lower levels of I-position than Jews. However, Arabs and Jews did not differ in their levels of emotional reactivity+fusion with others were related to the hypotheses assumed by authors”). No results related to hypotheses number 3 and 4 are presented on Abstract

Answer: Thanks to the reviewer's comment the results of these hypotheses have been added to the Abstract.

Comment:  Lines 367-370: The sentence “The present study presents a preliminary analysis of possible cultural differences in T2DM, DoS (I-position, emotional cutoff, and emotional reactivity+fusion with others), and emotional distress (anxiety and depressive symptoms). The aim was to investigate whether cultural variation in DoS and emotional distress indices may increase the risk of developing diabetes” was not related to the 4 hypotheses authors stated on Abstract.

Answer: The wording of the objectives has been corrected.

Comment: Lines 134-137: “In light of the disproportionate incidence of T2DM among ethnic minorities and specifically among Israeli Arabs, the study tests cultural and gender differences in DoS and emotional distress (anxiety and depressive symptoms) that might be associated with T2DM.”Taken into account the previous sentence, why hypotheses 2 and 3 focused on all participants instead of diabetes patients only?

Answer: The statistical analyzes were performed twice -once  for healthy and diabetic Jews and once for healthy and diabetic Arabs

Comment:  On lines 444-446: “The fourth htpothesis assumed associations between DoS and emotional distress (anxiety and depression). We looked at this issue comparing between ethnic groups in the indices that predict T2DM”. I can not understand the relation between the two previous sentences. Associations between DoS and emotional distress is different from the comparison between ethnic groups of the índices of Dos and emotional distress that can predict T2DM!!

Answer: Thanks to the reviewer's comment this sentence has been corrected. It is now written that: " the fourth hypothesis assumed associations between DoS and emotional distress (anxiety and depression). These relationships were examined separately for Jews and Arabs".

Minor Comments:

Comment: On Abstract, authors must add the purpose of the study before presenting the hypotheses. If the word limit does not allow i tis preferable to add the purpose of the study instead of the hypotheses.

Answer: The purposes have been added to the abstract instead of the hypotheses.

Comment 2: Differentiation of self is not the same concept as emotional cutoff and on Abstract both concepts are described as DoS.

Answer: Emotional cut-off is one of the dimensions of differentiation of self. Thanks to the reviewer's comment, a description of the dimensions of DoS has been added to the abstract.

Comment: References are not according to authors’guidelines. Authors must check it. In the text, reference numbers should be placed in square brackets [ ]; References must be numbered in order of appearance in the text.

Answers: References are now numbered.

Comment: Regarding authors’guidelines, authors must also chack the type and size of letter. For example, the letter on Bibliography is of a different size of the rest of the paper.

Answer: The type and size of letters have been changed.

Comment 5: On line 40, authors must change “sports” by “physical activity”

Answer: The word "sport" has been changed to "physical activity".

Comment: On line 48, authors must change “bodily glucose levels” by “glucose levels”

Answer: The word "bodily" was deleted.

Comment: Lines 185-186: Authors must change “glycemic control levels” by “blood glucose levels”.

Answer: The words " glycemic control levels " have been changed to " blood glucose levels ".

Comment: Another question I have: individuals with values between 100 mg/dl and 126 mg/dl are considered healthy or with diabetes?

Answer: Diabetic individuals diagnosed with T2DM (with fasting glucose) ≥126mg/Dl.

Comment: On line 245, authors stated that “ANCOVA was performed to assess ethnic group differences in anxiety and depressive symptoms.” As ANCOVA evaluates whether the means of a dependent variable are equal across levels of a categorical independent variable, I question authors if they use ANCOVA or MANCOVA to assess ethnic group differences in anxiety and depressive symptoms?

Answer: I performed ANCOVA since these 2 variables are entirely different indices. 

Comment: On line 402: Authors must change Looking at the two ethnic groups, the first hypothesis was corroborated. Arabs by “Regarding Hypothesis 2, Arabs”

Answer: This sentence has been corrected.

Sincerely yours
